# Tele-Assessment of Functional Capacity through the Six-Minute Walk Test in Patients with Diabetes Mellitus Type 2: Validity and Reliability of Repeated Measurements

**DOI:** 10.3390/s23031354

**Published:** 2023-01-25

**Authors:** Garyfallia Pepera, Evmorfia Karanasiou, Christina Blioumpa, Varsamo Antoniou, Konstantinos Kalatzis, Leonidas Lanaras, Ladislav Batalik

**Affiliations:** 1Clinical Exercise Physiology and Rehabilitation Laboratory, Department of Physiotherapy, Faculty of Health Sciences, University of Thessaly, GR-35100 Lamia, Greece; 2Independent Researcher, GR-35100 Lamia, Greece; 3Department of Internal Medicine, General Hospital of Lamia, GR-35100 Lamia, Greece; 4Department of Rehabilitation, University Hospital Brno, 62500 Brno, Czech Republic; 5Department of Public Health, Faculty of Medicine, Masaryk University, 62500 Brno, Czech Republic

**Keywords:** tele-assessment, remote assessment, telehealth assessment, functional capacity, diabetes mellitus, six-minute walk test

## Abstract

A tele-assessed 6MWT (TL 6MWT) could be an alternative method of evaluating functional capacity in patients with diabetes mellitus type 2 (DM2). This study aimed to assess the validity and reliability of a TL 6MWT. The functional capacity of 28 patients with DM2 (75% men) aged 61 ± 13 years was evaluated twice via an indoor, center-based 6MWT (CB 6MWT) and twice outside each patient’s home via a web-based platform TL 6MWT. The study showed a high statistically significant correlation between the CB and TL 6MWT (Pearson’s *r* = 0.76, *p* < 0.001). Reliability testing showed no statistically significant differences in the distance covered (CB1: 492 ± 84 m and CB2: 506 ± 86 m versus TL1: 534 ± 87 m and TL2: 542 ± 93 m, respectively) and in the best distance of the TL 6MWT (545 ± 93 m) compared to the best CB distance (521 ± 83 m). Strong internal reliability for both the CB (intraclass correlation coefficient (ICC) = 0.93) and the TL 6MWT (ICC = 0.98) was found. The results indicate that a TL 6MWT performed outdoors can be a highly valid and reliable tool to assess functional capacity in patients with DM2. No learning effect between the TL and CB assessment was found, minimizing the need for repetition.

## 1. Introduction

Functional capacity is a significant predictor of mortality and morbidity in patients with diabetes mellitus type 2 (DM2) [1]. Several studies have shown the reduced functional capacity of patients with DM2 compared with the healthy population [2,3]. Although peak oxygen uptake during cardiopulmonary exercise testing is the gold standard for evaluating aerobic functional capacity, it strains the patient’s circulatory system; thus, it is not recommended for patients with DM2 [4]. The measurement of maximal oxygen uptake is inappropriate in diabetic patients due to the potential risk of large muscle fatigue, increased blood pressure, decreased respiratory function, and hypoglycemia [4,5,6]. The six-minute walk test (6MWT) is a submaximal exercise test that contributes information related to functional capacity and efficiently indicates a patient’s ability to perform daily activities which are related to their quality of life [7,8]. Additionally, it may be a reliable tool for monitoring changes in performance over time in patients with DM2 [9]. Covering distances of less than 300 m is often associated with a poorer medical prognosis complicated by frailty, sarcopenia, or muscle mass loss [10,11].

The 6MWT has been conducted under various conditions, such as in straight or circular tracks of varying lengths, indoors or outdoors [5,12,13]. A comparison of indoor and outdoor testing revealed that the indoor 6MWT was comparable. While the current guidelines for the 6MWT recommend using a standardized 30m straight track with strict environmental control [14], it is therefore unknown whether this can be achieved in the home setting through tele-assessment. One negative effect of using track corridors shorter than 30 m is the fewer meters being covered due to time wasted while patients are turning [15,16,17]. In patients with low-performance levels, the length of the indoor track does not affect the outcome of the 6MWT [13]. In patients with chronic obstructive pulmonary disease, it has been found that a 6MWT performed outdoors on a 30 m track within reasonable climatic parameters could substitute for an equally long indoor 6MWT [12]. To date, a comparison of the distance achieved during a home versus center-based 6MWT has not been performed in patients with DM2.

Adsett et al. have shown that when the 6MWT is repeated at least twice, the final distance covered during the testing tends to go from 301 m in the first test to 313 m in the second test (*p* < 0.001) [18]. There is, therefore, a learning effect due to the familiarity with the test, and two repetitive measurements are enough for verification [19]. Home-based telemonitoring has proven to be a safe and effective alternative to center-based programs and can result in higher exercise compliance [20] and compliance with dietary recommendations in patients with DM2 [21]. Thus, telehealth could represent a valuable method to improve blood sugar control, prevent rehospitalization, enhance the quality of life, and improve adherence rates to therapeutic interventions in patients with DM2 [21,22,23]. Telehealth services also reduce costs and increase the potential to attend more sessions compared to center-based facilities [24]. Specifically, the COVID-19 pandemic dramatically increased the application of telehealth and the need for reliable remote monitoring of chronic patients [25]. Regardless, more research into home testing using the tele-assessed 6MWT (TL 6MWT) is necessary in order to evaluate its efficacy as an assessment tool for the functional capacity of DM2 patients. Several logistic and financial barriers limit access to health care. Thus, the implementation of tele-assessment could serve as a helpful alternative. If exercise testing is found to be accurate at home, it will reduce the need to visit hospitals for testing and further reduce the accessibility barriers for patients.

The primary aim of this study was to investigate, for the first time, the validity and reliability of repeated measurements of the 6MWT using the tele-assessment process in patients with DM2. The secondary aim was to identify if there is a learning effect due to the repetitive measurements in both assessment tests.

## 2. Methods

### 2.1. Study Design

The present study was a comparative study performed from the 9th of October to the 10th of November 2021 involving patients with DM2. The study was performed following the principles in the Declaration of Helsinki. The study protocol was approved by the Ethics Committee of the Department of Physiotherapy at the University of Thessaly (653/09-09-2021) and by the Scientific Council of the General Hospital of Lamia, Greece (Σ/21228).

### 2.2. Sample

Twenty-eight patients (21 male, aged 61 ± 13 years) volunteered and were eligible to participate in this study. Patients were recruited from the outpatient diabetes clinic of the public General Hospital of Lamia, a private practice of internal medicine specializing in diabetes in Lamia, and the Diabetic Foundation of Fthiotida. Eligibility criteria were (a) patients diagnosed with DM2 for at least 2 months and following the American Diabetes Association guidelines [26], (b) patients aged above 40, (c) patients who were under medical supervision and receiving pill or insulin medication for the control of their glucose levels, and (d) patients with an internet connection at home. Exclusion criteria were: (a) the presence of DM type I, (b) orthopedic or surgery associated disorders affecting walking ability, (c) unstable cardiovascular and musculoskeletal disease, and (d) communication difficulties. Study investigators checked potential patients for eligibility based on their medical records. Each patient was provided with an information sheet (explanation of the procedures, risks, and benefits of the study) and an informed consent form to be signed before participation. The order of testing for the eligible patients was random and determined by a computer application.

The sample size was calculated using the G*Power 3.1.9.4 program. The t-test (the difference between two dependent means and matched pairs) and the detection of a moderate effect size f = 0.6 (two-tailed, alpha level = 0.05, power = 80%) showed that a total of twenty-four patients were required. After adjustment, an estimated drop-out rate of 10%, the final total sample size was estimated at twenty-seven patients.

### 2.3. Data Collection

The assessor was a qualified physiotherapist in the use of the 6MWT before the start of the study testing. The same assessor carried out all assessments. After each assessment, the data were stored and presented in a spreadsheet for data analysis.

### 2.4. Assessment

Baseline characteristics were firstly assessed, including (a) anthropometric and demographic data (age, weight, and height), (b) medical/pharmacological history (insulin), and (c) hemodynamic data (glycated hemoglobin, heart rate, oxygen saturation, and blood pressure).

Patients undertook a 6MWT within two settings: (a) a standard center-based 6MWT assessment (CB 6MWT) and (b) a remote tele-assessed 6MWT (TL 6MWT). The CB 6MWT was conducted at the Department of Physiotherapy of the University of Thessaly, and the TL 6MWT was carried out outside each patient’s home. Two tests were conducted for each setting (center-based, CB; tele-assessment, TL), with the best test score used for comparison. All tests were performed simultaneously and within five days to reduce test heterogeneity due to patients’ different fitness levels/severity of DM2. Previous studies have shown similar reliability of the 6MWT with test–retest periods, ranging from one day (intraclass correlation coefficient (ICC) 0.93) to seven days (ICC 0.92) [27]. Each patient performed the two repetitions of the CB 6MWT with at least half an hour of rest between the tests. In the following five days, the patients had to undertake the two TL 6MWTs with the same half-hour interval between each. The same physiotherapist continuously supervised both the CB 6MWT and TL 6MWT (via video call) assessments.

The 6MWT was repeated twice to examine the reliability of both the CB and TL 6MWT and to show any learning effect that may result from performing the specific functional test.

#### 2.4.1. Center-Based 6MWT

The CB 6MWT was completed indoors on a level, temperature-controlled, straight track of 30 m in length on a hard surface from 8 to 10 a.m. The temperature was constant from 16 °C to 20 °C, following standardized guidelines [28]. For safety reasons, the assessor supervising the CB 6MWT was initially trained in immediate treatment if necessary.

All instructions and monitoring were consistent for the two tests. The primary measurement was the distance the patient could walk in six minutes. Blood pressure, heart rate, oxygen saturation (SpO_2_), and ratings of perceived exertion using the Borg (6–20) scale [29] were measured and recorded three times for each 6MWT (at the baseline, immediately after the completion, and one minute after the test). Heart rate and SpO_2_ were monitored using a portable pulse oximeter (Pulse Oximeter FOX-350, I-TECH, Italy), and blood pressure was monitored with an automated arm blood pressure monitor (BP-B2 Easy, Microlife Corporation, Switzerland) [30]. Preparation for patients included wearing comfortable clothes and shoes suitable for walking. Patients were advised to have a light meal before the test and avoid coffee and smoking two hours before the testing procedures [28]. Before beginning the testing, the physiotherapist gave the necessary verbal instructions to all patients for the 6MWT procedure. When the patients were ready, they stood at the starting position next to the cone. At the physiotherapist’s command, they started walking, and a countdown of six minutes started. The number of laps, the distance walked, and the number of turns performed by the patients were counted, and the length of the track was recorded.

The 6MWT was terminated if chest pain, severe shortness of breath, leg cramps, tremors, sweating, and paleness appeared. The hemodynamic parameters were continuously evaluated through the blood pressure monitor and the pulse oximeter.

#### 2.4.2. Tele-Assessment of the 6MWT

The TL 6MWT was always performed outdoors in the presence of a familiar patient in a 30 m outdoor corridor. Each patient was located in an outdoor area of their home and communicated with the physiotherapist remotely. To enable the performance of this test, the patient positioned the mobile phone at the start of the six-minute walk track to allow the assessor to monitor the whole 30 m track. The 6MWT was conducted when the environmental conditions were suitable (a temperature between 16 °C and 20 °C, no rain, a wind speed of less than 20 km/h, and 60–70% humidity, according to the Greek National Meteorological Service). Additionally, a hard, level surface had to be available, and any noise or distractions likely to affect walking performance had to be minimized [12]. A 30 m straight-shaped corridor was required to resemble the indoor CB 6MWT [13,31,32]. The evaluator assessed the suitability of the external environmental conditions before the 6MWT. The TL 6MWT track had to be as similar as possible to the CB for length and shape, prioritizing track shape (straight) over the track length.

Before the TL 6MWT, there was a detailed written and oral presentation of the equipment patients had to use. The trial use during the CB 6MWT helped patients become familiar with the measurement process. More specifically, the patients knew that before the 6MWT, they had to rest for 10 min and then have their blood pressure measured.

During the TL 6MWT, patients measured their blood pressure, heart rate, and SpO_2_ using a micro life or Omron electronic sphygmomanometer following the assessor’s recommendation. Blood pressure, heart rate, SpO_2_, and fatigue were measured and recorded three times for each trial of each 6MWT. In addition, a person from the patient’s family was necessary to hold a mobile phone, repeat the physiotherapist’s phrases during the 6MWT, and help monitor and verbally report the patient’s vital measurements. The examiner counted the number of laps completed during the 6MWT and gave standardized encouragement per recommended guidelines [33]. The patient’s family member also used a lap counter to confirm the number of laps achieved during the test. When the test was completed, the patients were asked to read the nearest distance marker on the track. The physiotherapist recorded a 6MWT distance to the nearest meter.

The telecommunication connection was achieved using smartphones with a direct video call through Skype. A test call via Skype was made several days before the final TL 6MWT.

As for the safety of the tele-assessment, the physiotherapist asked for the presence of a familiar person to offer help during any discomfort. In addition, the patients were given written support material with instructions for measuring blood pressure, heart rate, and SpO_2_.

### 2.5. Statistical Analysis

Statistical analysis was carried out using Statistical Package for the Social Sciences Version 25. Data are presented as means and standard deviations.

The differences between the two 6MWT measurements in each condition (center-based, CB and tele-assessment, TL) were evaluated using the paired t-test (for *p* < 0.05). Statistical analysis included dispersion analysis and was then validated via Pearson correlation for the 6MWT between the CB and TL assessment (for *p* < 0.05). For the evaluation of the difference in distance between the two conditions, statistics compared the test with the best distance score for each condition. Comparisons were made for heart rate, fatigue, SpO_2_, and arterial pressure between the two conditions for all assessment points (before, after, and recovery). The minimum clinically significant difference for the 6MWT was 27 m [9].

The ICC was used to evaluate the reliability of the 6MWT repeated measurements between the two TL tests and the two CB tests. Agreement for the 6MWT distance between two TL tests was examined using the Bland–Altman method [34]. Interpretation of the ICC represents values of less than 0.5 indicating poor reliability; values between 0.5 and 0.75, indicating moderate reliability; values between 0.75 and 0.9, indicating good reliability; and values greater than 0.9, indicating excellent reliability.

## 3. Results

A total of 28 patients were included in this study. All examinations were completed by 23 patients. The flow of the study is shown in Figure 1. The mean age of patients was 61 ± 13 years. Of the cohort, 75% of the patients were male. The patients participated in moderate physical activity after presenting a mean of 2.750 MET minutes per week. In the study group, 43% and 57% of the sample received insulin and pills, respectively (Table 1). None of the patients used walking aids. For the TL 6MWT, the mean temperature was 18 degrees Celsius, and the wind speed was 23.5 km/h, all considered perfect conditions.

Figure 2 illustrates the mean difference and limits of agreement for the two TL 6MWTs (TL1 and TL2). TL1 was presented as 534 ± 87 m, while TL2 was 542 ± 93 m (*p* = 0.14). There was no significant increase in distance between the two TL 6MWT. Using the Bland–Altman method between the TL measurements, there was a small mean difference on the retest of 8 m within 95% agreement limits from −38 m to 53 m. Similarly, for the CB 6MWT, there was no significant statistical increase in distance (14 m, *p* > 0.05) with patients walking further on the second test. The test–retest reliability for the CB 6MWT was high with an ICC of 0.93 (95% CI 0.85–0.97). Reliability for the TL 6MWT was high with an ICC of 0.98 (95% CI 0.95–0.99).

The differences between the CB and TL 6MWT were examined using paired t-tests for all variables. The 6MWT using both CB and TL ranged from 492 to 714 m. The validity test showed a high statistically significant correlation for the 6MWT between the CB and TL 6MWT (Pearson’s r = 0.76 and *p* < 0.001) (Figure 3). A non-statistically significant mean difference of 24.4m (*p* = 0.07) was found between the CB and TL 6MWTs (Table 2). Between the two conditions, comparisons revealed a significant difference in the heart rate, blood pressure, and fatigue values in the CB 6MWT. No adverse events were reported during overall (n = 98) 6MWT measurements.

## 4. Discussion

The present study is the first to investigate the use of TL assessment to evaluate functional capacity in patients with DM2 via the 6MWT. The results of this study demonstrate that the TL 6MWT using a standardized protocol in an outdoor setting is safe and can be similar to the standard indoor CB 6MWT. More specifically, our study confirmed the validity and reliability of using TL assessment to administer the 6MWT by presenting excellent reliability values in the CB and TL 6MWT assessments (ICC > 0.9).

When determining the validity of the TL 6MWT, our results have demonstrated a non-significant difference in the final meters covered in favor of the TL 6MWT. This small difference may be due to the fact that the sample’s initial hemodynamic characteristics were better in the TL 6MWT, thus explaining the better performance accomplished during the test. Between the two conditions, comparisons revealed an increase in heart rate, blood pressure, and fatigue in the CB 6MWT. It is due to certain stressful factors during the whole process of the CB 6MWT. While patients were in their home environment during the TL 6MWT, they presented lower values of the vital measurements due to the fact that they knew the entire test process and the setting was familiar. Furthermore, during the TL 6MWT, SpO_2_ values showed an improvement with higher values because the tests were performed outside, which could result in better breathing.

However, no similar studies in DM2 populations have evaluated the validity and reliability of a TL 6MWT. Previous studies in different patient populations display the TL tools as valid and reliable compared to CB assessment ones [35,36]. For example, in patients with cystic fibrosis, there were no significant differences in oxygen saturation, heart rate, and rate of perceived exertion between TL and CB assessments for the three-minute step test [35,37]. Similarly, Hwang et al. (2016) presented excellent reliability results for the TL 6MWT (ICC > 0.95) in assessing functional capacity in patients with chronic heart failure [36]. It is worth mentioning that assessors carried out the TL assessment in the same building. However, in a separate room, it may have positively affected the patient’s self-confidence and feeling of safety, thus leading to favorable results. On the contrary, the study by Holland et al. (2015), in which a remote TL 6MWT was performed in any available corridor (<30 m) present at home, either indoors or outdoors, the 6MWT did not validly assess the performance of patients, mainly due to the short length of the available indoor corridors (usually 17 m) [38].

Previous research has been limited to CB assessment of functional capacity in diabetic patients using the 6MWT and other exercise tests [3,11,39,40]. The 6MWT is a suitable tool for assessing functional capacity in diabetic patients due to its good acceptability and its easy incorporation into daily living activities; thus appearing as the best among the current walking tests [8]. A large number of studies have reported the necessity and utility of telehealth through various applications for patients with DM2, including telelearning, telemonitoring, DM self-management, and telecounseling [21,24,41,42]. Telemedicine studies, such as the research by Milani [23], have shown that telehealth applications can increase glycemic control, reduce glycated hemoglobin values, and improve the quality of life [43,44,45]. Regarding the implementation of telerehabilitation for diabetic patients, only one study by Duruturk (2019) has investigated the effect of a remote intervention in glucose control, physical activity, muscle strength, and patient’s psychosocial status [46]. Several studies propose telerehabilitation interventions as safe, effective, alternative, and more cost-effective ways of providing secondary prevention for the diabetic population [46,47].

It is worth noting that wise the CB 6MWT, a 30 m corridor was also used during the TL 6MWT. When the 6MWT is performed outdoors in a 30 m corridor, with reasonable climatic parameters, it could equal the performance of the 6MWT indoors in a 30 m corridor [12]. Furthermore, a 30m corridor was chosen since the length is a critical factor in the distance covered during the 6MWT. It has been found that a corridor shorter than 30 m requires patients to spend more time on turns [44]. However, unrelated to the number of turns, other factors may influence the distance covered during a 6MWT. Higher gait speed with greater stride velocity was used over longer tracks (greater than 20 m) when walking at habitual speed, whilst a slower speed was adopted on shorter track (less than 10 m) by healthy elderly subjects [48,49]. In addition, changes in gait strategy may be adopted on a shorter track length, resulting in a reduced 6MWT distance [50]. Therefore, tracks of longer length permit a more significant amount of acceleration and a higher top speed [51]. Furthermore, this would probably suit the diabetic population most as shorter distances no longer show sufficient validity [19].

Although several studies have shown a learning effect during successive repetitions of the 6MWT [13,52,53], no learning effect was observed between the tests in the CB or TL in the present study. This could have been caused by the short rest period (30 min) between the two repeated measurements. In addition, Spencer et al. (2008) claimed that no significant differences were noted when repeated 6MWT measurements were taken immediately on the same day after a pulmonary rehabilitation program [19]. However, there are conflicting views on this issue [38]; therefore, further research with a larger group will be needed to confirm the learning effect.

Regarding the safety of the TL 6MWT, a thorough pre-test screening and recording of the participants’ medical history and physical status and the presence of a familiar person during the TL procedure would minimize the incidence of any adverse events [13,54,55]. Additionally, real-time monitoring of the implementation of the TL 6MWT via web-based platforms by medically qualified staff would further enhance the TL 6 MWT safety.

### 4.1. Strengths and Limitations

Despite the originality of this study and the favorable results, several limitations need to be mentioned. Difficulties related to technical issues included poor outdoor Wi-Fi internet connection availability, which led to patients needing mobile data. The involvement of a family member during the TL required extra communication. Artificial intelligence-based applications in the future could solve this barrier.

In addition, patients used different types of telemonitoring and evaluation equipment. For example, a blood pressure monitor and a pulse oximeter could have resulted in an assessment bias recall. Lastly, the origin of the study sample being from a provincial town and the small representation of women in the sample reduce the potentiality of generalization of the results to the whole population.

### 4.2. Practical Implications and Future Perspectives

The study showed that a remote TL 6MWT assessed outdoors in reasonable climate parameters can reflect a CB 6MWT indoor physical fitness assessment. The nature of the outdoor TL 6MWT is more reflective of real life compared to the conventional 6MWT, where the artificial corridor environment is usually free of distractions. Hence an outdoor TL 6MWT could be an alternative measurement to conventional CB 6MWT in patients with DM2 for use in practice, especially when an indoor corridor is unavailable.

Furthermore, the results of this study will allow researchers and/or health care professionals to use a conventional physical fitness CB 6MWT to predict outdoor physical fitness. In addition, in the future the use of the TL 6MWT may allow healthcare professionals to better assess the rural patients’ physical fitness and prescribe and monitor the effectiveness of individualized exercise activities which can optimize both the treatment outcomes and the utilization of preventive and rehabilitation interventions.

Future research should focus on studying how TL can be affected by different outdoor conditions, such as different terrains, air pollution, and weather (summer and winter temperatures and wind), and how these affect the physical performance of patients for more generalizing the use of remote TL 6MWT.

Finally, using reliable and valid wearable sensors, such as sports watches, may contribute to a better and more objective assessment of a patient’s hemodynamic characteristics during a TL 6MWT. The implementation of wearable-assisted TL in the future may play a vital role in the widespread provision of adequate medical care, even under unfavorable socioeconomic conditions, to a more significant part of the patient population.

## 5. Conclusions

In conclusion, the data indicate that a TL 6MWT performed outdoors can be a reliable and valid tool for assessing functional capacity in patients with DM2. There is no learning effect between the two tests in both the TL and CB 6MWTs, dismissing the need for second tests. Future studies are to be carried out with larger and more representative samples to increase the generalizability of these findings.

## Figures and Tables

**Figure 1 sensors-23-01354-f001:**
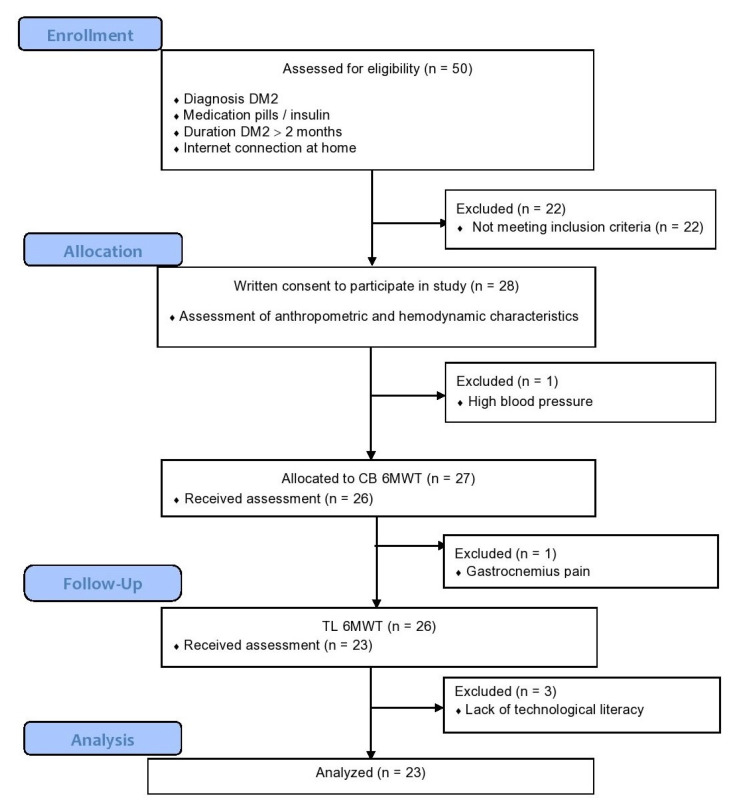
Flowchart of the study.

**Figure 2 sensors-23-01354-f002:**
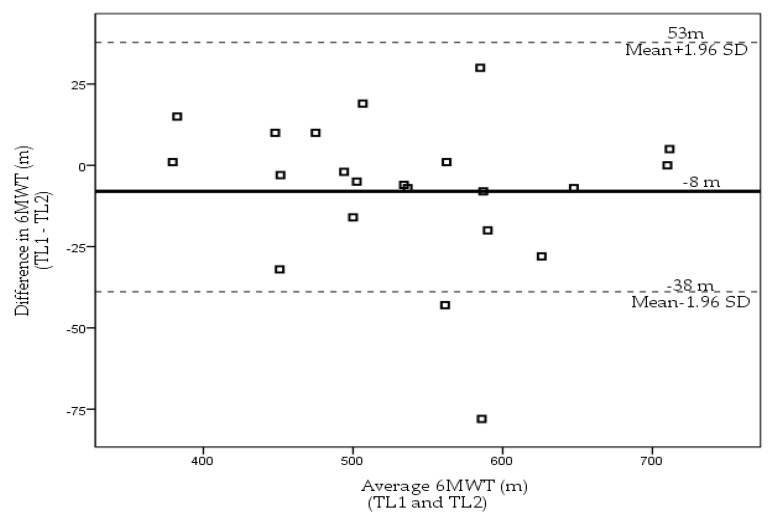
Limits of agreement plot (Bland–Altman Plot). Intra-individual differences between performances on the 6MWT on the two TL 6MWTs (from TL1 to TL2) plotted against intra-individual average distance scores of the TL1 and TL2. The central line represents the mean of the intra-individual differences, and the flanking lines represent the 95% limits of agreement. Mean difference is −8 m.

**Figure 3 sensors-23-01354-f003:**
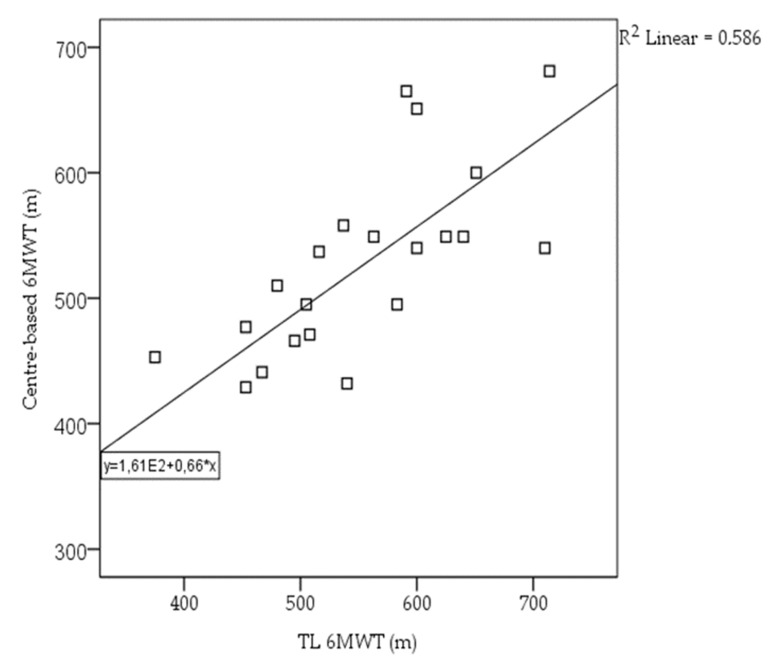
Correlation between the CB and TL 6MWT.

**Table 1 sensors-23-01354-t001:** Patients’ demographic and baseline characteristics.

Variables	Mean ± SD	Range
Anthropometric Characteristics		
age (years)	61 ±13	39–85
gender (N %)	21 males (75%) 7 females (25%)	
medications (N %)	12 (42.9%) insulin 16 (57.1%) pills	
weight (kg)	93.1 ± 21.0	65–152
height (cm)	159.5 ± 45.5	150–189
BMI (kg/m^2^)	31.8 ± 7.6	25–60
waist size (cm)	111.1 ± 14.3	88–142

HR: heart rate; SD: standard deviation of the mean; BMI: body mass index.

**Table 2 sensors-23-01354-t002:** Evaluation of the differences of the CB and TL 6MWT assessment.

Variable	CB 6MWT	TL 6MWT	Difference	*t*-Test Value
6MWT (best score) (m)	520.5 ± 80.3	544.8 ± 93.1	−24.3 ± 60.6	−1.8
HR pre-6MWT (bpm)	81.2 ± 14.5	77.9 ± 12.6	3.2 ± 11.7	1.2
HR end of the 6MWT (bpm)	100.8 ± 20.7	86.8 ± 16.1	14.0 ± 20.0	2.9 ^a^
HR 1-min post-6MWT	86.5 ± 15.8	83.3 ± 12.3	3.2 ± 12.0	1.2
RPE end of the 6MWT	7.6 ± 2.8	7.3 ± 2.5	0.2 ± 1.1	1.0
SpO2 pre-6MWT (%)	97.4 ± 1.9	97.7 ± 1.3	−0.3 ± 1.7	−0.9
SpO2 end of the 6MWT (%)	97.2 ± 1.9	97.8 ± 1.9	−0.6 ± 1.5	−1.6
SBPr pre-6MWT (mmHg)	138.7 ± 15.4	131.0 ± 15.6	7.7 ± 14.4	2.3 ^a^
SBP end of the 6MWT (mmHg)	151.3 ± 23.9	144.8 ± 24.2	6.4 ± 15.7	1.7
DBP pre-6MWT (mmHg)	86.5 ± 10.9	80.6 ± 14.0	5.8 ± 9.7	2.6 ^a^
DBP end of the 6MWT (mmHg)	92.1 ± 10.5	86.2 ± 17.0	5.8 ± 15.2	1.6

^a^ indicate significance (*p* < 0.05); values are expressed as mean and standard deviation. CB, center-based; TL, tele-assessment; 6MWT, six-minute walk test; HR, heart rate; RPE, rate of perceived exertion scale (1–10); SpO_2_, oxygen saturation; SBP, systolic blood pressure; DBP, diastolic blood pressure.

## Data Availability

Not applicable.

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
