# Peer review of "Tele-Assessment of Functional Capacity through the Six-Minute Walk Test in Patients with Diabetes Mellitus Type 2: Validity and Reliability of Repeated Measurements"

_sensors, 2023, doi:10.3390/s23031354_

Round 1

Reviewer 1 Report

Interesting, well-designed study on a relevant topic in the present times. Limitations of the study should be acknowledged, particularly related to the absence of wearables which could have strengthened the relevance of the study.

Author Response

Dear reviewer,

I appreciate the time you took to review. Thank you for your suggestions. They have been addressed.

Reviewer 2 Report

Dear authors, 
Although well-written and structured, the manuscript could be improved with major changes. The suggestions are described below. 

Title
The title is appropriate but could include whether the sample was composed of adults or elderly subjects and mention some possible comorbidities (i.e., obesity, hypertension, etc.).

Abstract
Regarding the description of the objective, I suggest removing the learning effect. 
The sentence (lines 16-18) needs to be revised, especially the punctuation. 
Regarding the TL, more information is needed. Then, I strongly recommend revising the whole abstract to ensure a clarification about the TL.  When reading the manuscript, it is clear that two tests were performed but the abstract requires this information as well. The same applied to other methodological information.
It could be interesting to mention the statistical analysis made for each purpose and then describe the results. In addition to the main findings, the authors should provide some perspectives or practical implications at the end of the manuscript. 

Introduction

The authors need to revise the first paragraph. I disagree with the adopted approach suggesting that CPET is inappropriate for patients with DM2. The authors should emphasize the advantages and disadvantages of using CPET and field walking tests. Then, the next paragraph introduces the 6MWT as a reliable and practical measurement of functional capacity. It is important to add citations for the validation of this test for different populations, especially those with chronic conditions.

The second and third paragraphs could be revised and turned into only one. I also suggest citing one or more references for the comparison between indoor and outdoor testing for more in-depth consideration.

The introduction needs to clearly set the relevance of home-based testing, as well as the role of teleassessment. It is also important to distinguish between them. For instance, it is a synchronous strategy or a self-instructed test using a smartphone app, among others.

The fourth and fifth paragraphs seemed more like a possible discussion than an introduction. I recommend revising them.

Regarding telemedicine, the authors need to make it more clear whether it is referring to telemonitoring or telemedicine. It also could be more interesting to use telehealth instead of telemedicine. Since it is a key point from the study, the authors could include m-health, e-health, synchronous and asynchronous, and what is already known for this approach to subjects with chronic conditions. The introduction would be enhanced with advantages, disadvantages, limitations, and practical implications of telehealth for subjects with chronic conditions, especially DM2.   

Methods

I only recommend revising to check typing or punctuation errors.

Table 1 is a characterization of the study sample and should be presented in the results section following the flowchart.

Replace profession with occupation. 

The methods section mentioned assessments that do not have variables presented in the results. This must be revised for adding this information. Otherwise, it would be better to exclude this description. Additionally, how the authors obtained the variables in Table 1 must be described. It was a previous assessment or this study is a part of a larger study, among others. 

In my opinion, it is unclear if all CB tests were conducted prior to the TL tests. If this is correct, it is a potential bias. Then, I am looking forward to clarification.

I also recommend that the tests could be part of a distinct topic from the other assessments. This change helps the reader to understand what the CB and TL tests were alike and the main differences in execution etc.

 For both CB and TL, I suggest describing them in chronological order. This description could help the reader to understand the application of the standardized version of 6MWT as recommended by ATS and how the test was performed in the other context (i.e. TL). 

For instance, the communication with the physiotherapist occurred throughout the test or before? How were the conditions of the outdoor corridor? The physiotherapist only checked the corridor through video. What time or period were performed the tests? Was the surface plane, especially when taking into account Greece's geographical characteristics?

Lines 178-183: This information needs to be placed in the introduction or the discussion. In the methods, the authors should only describe/report the study protocol instead of justifying the methodology.  

The same chronological order description applies to the TL test. 

The statistical analysis was well-written and needs only a minor revision for checking typing or punctuation errors.

Results

Table 1 should be presented here. 

The description in some sentences is repetitive. Thus, I recommend revising the whole section. For instance, I suggest characterizing the patients as middle-aged subjects instead of repeating the mean and standard deviation, which by the way are different in table and text (60.8 ± 12.6 and 61 ±13, respectively). Once again, the results mention the quality of life, but this data should be also presented in table 1. Similarly, the methods need the aforementioned changes. 

For the tests, I suggest describing them in a new paragraph.  

I also have to point out that the flow chart was not mentioned in the text. The authors need to revise it carefully.

It would be better to report results for CB and then TL.

In my opinion, the topics in the results are unnecessary. I recommend removing them.  

Discussion

The authors must be cautious when affirming the absence of similar studies. I recommend reading this recent paper: https://pubmed.ncbi.nlm.nih.gov/35162141/

The second paragraph seems a summarized version of the results. This must be improved. 

The discussion requires a major and careful revision. The authors were able to discuss relevant aspects, but, at the same time, the discussion seemed disconnected from the results of the present study. 

Indeed, discussing corridor length is needed. I agree with that, but the approach was not adequate. The authors could discuss this aspect by referring as a strength of the study since the authors proposed a strategy to cope with potential bias or limitations regarding corridor conditions.

I suggest revising the text by discussing the main findings according to their relevance to this study. I also recommend emphasizing the results that fulfill your purpose. Lastly, I would like the authors to point out the major implications of using TL 6MWT, including how to ensure patient safety and how to screen patients able to perform this test (i.e., patients who (a) have a responsible familiar available for being the examinator, (b) are considered stable, (c) do not use walk aids, (d) have access to the internet with a minimum quality connection for a synchronous meeting, (e) understand the orientations for ensuring safety measures, etc.).    

I appreciate the topic's limitations. But I strongly recommend the authors change it into Strengths and limitations. Therefore, the authors should point out strengths and/or practical implications regarding this study.

Conclusion

I suggest better describing the sample. It is also interesting to revise the whole section for minor changes to ensure clarity and coherence among the different sections. 

Tables and figures

Table 1 should be presented in results instead of methods. 

The figures are small and the definition could be improved. I also recommend revising the footnote in figure 3. 

Table 2 should be revised. I suggest the following changes:
- before and after would be better presented if side by side;

- the authors could also avoid using acronyms since the table is relatively small;

- in my opinion, there is no need to present a p-value. The authors could use a symbol to indicate the significance;

- if the authors maintain this presentation, an alternative is mentioning heart rate with the unit measure, for instance, in italic on a line and following just with Before, After, and Recovery.

References

In this section, there is an error. The 7 is placed between 5 and 6.  

Author Response

Dear reviewer,

Thank you for taking the time to give us your valuable feedback. It is much appreciated.

Comment 1. Title
The title is appropriate but could include whether the sample was composed of adults or elderly subjects and mention some possible comorbidities (i.e., obesity, hypertension, etc.).

Reply. Thank you for your comment. Though we think that the title should not be changed since there was a wide range of age, from adults to elderly subjects, in the sample and no sample categorization based on any comorbid.

Abstract
Comment 1. Regarding the description of the objective, I suggest removing the learning effect.

Reply. Thank you for your comment. It was removed.

Comment 2. The sentence (lines 16-18) needs to be revised, especially the punctuation. 
Regarding the TL, more information is needed. Then, I strongly recommend revising the whole abstract to ensure a clarification about the TL.  When reading the manuscript, it is clear that two tests were performed but the abstract requires this information as well. The same applied to other methodological information.
It could be interesting to mention the statistical analysis made for each purpose and then describe the results. In addition to the main findings, the authors should provide some perspectives or practical implications at the end of the manuscript. 

Reply. Thank you for your comments. They were taken into account

Introduction

Comment 1. The authors need to revise the first paragraph. I disagree with the adopted approach suggesting that CPET is inappropriate for patients with DM2. The authors should emphasize the advantages and disadvantages of using CPET and field walking tests. Then, the next paragraph introduces the 6MWT as a reliable and practical measurement of functional capacity. It is important to add citations for the validation of this test for different populations, especially those with chronic conditions.

Comment 2. The second and third paragraphs could be revised and turned into only one. I also suggest citing one or more references for the comparison between indoor and outdoor testing for more in-depth consideration.

Comment 3. The introduction needs to clearly set the relevance of home-based testing, as well as the role of teleassessment. It is also important to distinguish between them. For instance, it is a synchronous strategy or a self-instructed test using a smartphone app, among others.

Comment 4. The fourth and fifth paragraphs seemed more like a possible discussion than an introduction. I recommend revising them.

Reply. Thank you for your comments. They were taken into account

Comment 5. Regarding telemedicine, the authors need to make it more clear whether it is referring to telemonitoring or telemedicine. It also could be more interesting to use telehealth instead of telemedicine. Since it is a key point from the study, the authors could include m-health, e-health, synchronous and asynchronous, and what is already known for this approach to subjects with chronic conditions. The introduction would be enhanced with advantages, disadvantages, limitations, and practical implications of telehealth for subjects with chronic conditions, especially DM2.

   Reply. Thank you for your comments. They were taken into account

Methods

Comment 1. I only recommend revising to check typing or punctuation errors.

Reply. Thank you for your comment. It was taken into account

Comment 2 Table 1 is a characterization of the study sample and should be presented in the results section following the flowchart.

Reply. Thank you for your comment. It was taken into account

Comment 3  Replace profession with occupation. 

Reply. Thank you for your comment. It was replaced

Comment 4 The methods section mentioned assessments that do not have variables presented in the results. This must be revised for adding this information. Otherwise, it would be better to exclude this description. Additionally, how the authors obtained the variables in Table 1 must be described. It was a previous assessment or this study is a part of a larger study, among others. 

Reply. Thank you. It was revised.

Comment 5  In my opinion, it is unclear if all CB tests were conducted prior to the TL tests. If this is correct, it is a potential bias. Then, I am looking forward to clarification.

Comment 6 I also recommend that the tests could be part of a distinct topic from the other assessments. This change helps the reader to understand what the CB and TL tests were alike and the main differences in execution etc.

Reply. Thank you for your comment. Though, we think that the CB and TL are being described in different sections.

Comment 7  For both CB and TL, I suggest describing them in chronological order. This description could help the reader to understand the application of the standardized version of 6MWT as recommended by ATS and how the test was performed in the other context (i.e. TL). 

Comment 1 For instance, the communication with the physiotherapist occurred throughout the test or before? How were the conditions of the outdoor corridor? The physiotherapist only checked the corridor through video. What time or period were performed the tests? Was the surface plane, especially when taking into account Greece's geographical characteristics?

Reply. Thank you for all your valuable comments. The physio checked the test only throught video.

The CB 6MWT was completed indoors on a level, temperature-controlled, straight track of 30 m in length on a hard surface from 8 - 10 a.m. The temperature was constant  from 16ο - 20οC, in accordance with standardized guidelines[28]. For safety reasons, the assessor supervising the CB 6MWT was initially trained in immediate treatment if necessary.

Comment 1 Lines 178-183: This information needs to be placed in the introduction or the discussion. In the methods, the authors should only describe/report the study protocol instead of justifying the methodology.

 Reply. Thank you for your comment. It was taken into account

Comment 1 The same chronological order description applies to the TL test. 

Comment 1 The statistical analysis was well-written and needs only a minor revision for checking typing or punctuation errors.

Results

Comment 1. Table 1 should be presented here. 

Reply. Thank you for your comment. Table 1 is mentioned in the results section.

Comment 2. The description in some sentences is repetitive. Thus, I recommend revising the whole section. For instance, I suggest characterizing the patients as middle-aged subjects instead of repeating the mean and standard deviation, which by the way are different in table and text (60.8 ± 12.6 and 61 ±13, respectively). Once again, the results mention the quality of life, but this data should be also presented in table 1. Similarly, the methods need the aforementioned changes. 

Reply. Thank you for your comments. They were taken into account.

Comment 3. For the tests, I suggest describing them in a new paragraph.  

Comment 4. I also have to point out that the flow chart was not mentioned in the text. The authors need to revise it carefully.

Reply. Thank you for your comment. Though, the flow chart is mentioned in the first paragraph of the results section.

It would be better to report results for CB and then TL.

In my opinion, the topics in the results are unnecessary. I recommend removing them.  

Discussion

Comment 1. The authors must be cautious when affirming the absence of similar studies. I recommend reading this recent paper: https://pubmed.ncbi.nlm.nih.gov/35162141/

Reply. Thank you for your comment. Though, our study was the only one implementing a remote, supervised TL 6MWT in DM2 patients and not in healthy population like the one used as sample in the aforementioned paper.

Comment 2.The second paragraph seems a summarized version of the results. This must be improved. 

Reply. Thank you for your comment. Though, we think that it is not a summarized version of the results, rather an explanation and discussion over the results found.

Comment 3. The discussion requires a major and careful revision. The authors were able to discuss relevant aspects, but, at the same time, the discussion seemed disconnected from the results of the present study. 

Reply. Thank you for your comment. It was taken into account.

Comment 4. Indeed, discussing corridor length is needed. I agree with that, but the approach was not adequate. The authors could discuss this aspect by referring as a strength of the study since the authors proposed a strategy to cope with potential bias or limitations regarding corridor conditions.

Comment 5. I suggest revising the text by discussing the main findings according to their relevance to this study. I also recommend emphasizing the results that fulfill your purpose. Lastly, I would like the authors to point out the major implications of using TL 6MWT, including how to ensure patient safety and how to screen patients able to perform this test (i.e., patients who (a) have a responsible familiar available for being the examinator, (b) are considered stable, (c) do not use walk aids, (d) have access to the internet with a minimum quality connection for a synchronous meeting, (e) understand the orientations for ensuring safety measures, etc.).

 Reply. Thank for your comment. It was taken under consideration.     

I appreciate the topic's limitations. But I strongly recommend the authors change it into Strengths and limitations. Therefore, the authors should point out strengths and/or practical implications regarding this study.

Conclusion

I suggest better describing the sample. It is also interesting to revise the whole section for minor changes to ensure clarity and coherence among the different sections. 

Tables and figures

Table 1 should be presented in results instead of methods. 

The figures are small and the definition could be improved. I also recommend revising the footnote in figure 3. 

Reply. Thank for your comment. They were revised. Footnote in figure 3 was revised.

Table 2 should be revised. I suggest the following changes:
- before and after would be better presented if side by side;

- the authors could also avoid using acronyms since the table is relatively small;

- in my opinion, there is no need to present a p-value. The authors could use a symbol to indicate the significance;

- if the authors maintain this presentation, an alternative is mentioning heart rate with the unit measure, for instance, in italic on a line and following just with Before, After, and Recovery.

Reply. Thank for your comments. Table 2 was formatted.

References

In this section, there is an error. The 7 is placed between 5 and 6. 

Reply. Thank for your comment. Though 7 applies to a page to the aforementioned 5th reference and not to a new reference. 

Reviewer 3 Report

The authors aimed to investigate, for the first time, the validity and reliability of repeated measurements of 6MWT through the TL assessment process in patients with DM2. A further aim was to identify if there is a learning effect between the repetitive measurements in both assessment tests.

The study covers some issues that have been overlooked in other similar topics. The structure of the manuscript appears adequate and well divided in the sections. Moreover, the study is easy to follow, but some issues should be improved. Some of the comments that would improve the overall quality of the study are:

a.      Authors must pay attention to the technical terms acronyms they used in the text.

b.    Conclusion Section: please add some "take-home message".

Author Response

Dear reviewer,

Thank you for this valuable feedback.

Comment 1. Authors must pay attention to the technical terms acronyms they used in the text.

Reply. Thank you for your comment. Some acronyms were corrected

Comment 2. Conclusion Section: please add some "take-home message".

Reply. Thank you for your comment. A "take-home message" was added.

Reviewer 4 Report

Dear Authors

I have reviewed your paper with great interest.

I will accept your paper after a minimal revision.

My revision is:

Title: Very Good

Abstract: Very Good

Introduction and AIM: The problem and the aim are well descripting.

Materials, Patients and methods and statistics: All good.

Results: Focus on and well described.

Discussion and Thread: effectiveness Focus ON.

The assessment of outcomes in the ankle fractures or in normal patient , they me be relevated by foot loading and gait analysis, please cite and discuss this paper:

Falzarano G, Pica G, Medici A, Rollo G, Bisaccia M, Cioffi R, Pavone M, Meccariello L. Foot Loading and Gait Analysis Evaluation of Nonarticular Tibial Pilon Fracture: A Comparison of Three Surgical Techniques. J Foot Ankle Surg. 2018 Sep-Oct;57(5):894-898. doi: 10.1053/j.jfas.2018.03.025. 

and to prevent skin problem in dyabete in fracture ankle, cite and discuss this paper:

Petruccelli R, Bisaccia M, Rinonapoli G, Rollo G, Meccariello L, Falzarano G, Ceccarini P, Bisaccia O, Giaracuni M, Caraffa A. Tubular vs Profile Plate in Peroneal or Bimalleolar Fractures: is There a Real Difference in Skin Complication? A Retrospective Study in Three Level I Trauma Center. Med Arch. 2017 Aug;71(4):265-269. doi: 10.5455/medarh.2017.71.265-269. 

Vitamic C reduces the inflammation of ankle:
Ripani U, Manzarbeitia-Arroba P, Guijarro-Leo S, Urrutia-Graña J, De Masi-De Luca A. Vitamin C May Help to Reduce the Knee's Arthritic Symptoms. Outcomes Assessment of Nutriceutical Therapy. Med Arch. 2019 Jun;73(3):173-177. doi: 10.5455/medarh.2019.73.173-177. PMID: 31404121; PMCID: PMC6643354.

References: Well chosen but to improve

Figures and Table: Very Good.

Author Response

Dear reviewer,

Thank you for your comments and your effort to review our manuscript. References are interesting but not relevant to the topic.

Round 2

Reviewer 2 Report

First of all, I appreciate the changes throughout the manuscript. There was a significant improvement, as well as a more readable version of this study for different types of readers. For me, this is essential in order to ensure a potential translation of this knowledge into a real practice scenario. 

Despite several of my suggestions were not taken into account, the overall comments were expectedly attended. Therefore, I recommend the publication since it is suitable for its main purpose and the journal's scope. However, some minor revision is required as the following comments:

Introduction

- Please, merge the second and third paragraphs into only one;

- Please, add information about the ceiling effect as well in the paragraph about the learning effect and made the necessary changes to ensure clarity;

- Please, merge the two last paragraphs into only one. If possible, leave the primary and secondary aims in a new paragraph.

Methods

- Lines 115-117: Were these variables collected independently from the study or not? Who collected them and when? It is also important to describe how these variables were achieved. 

Results

- Please, double-check the previous comments.

Table 2

- Similar to HR, provide the necessary changes in table 2 for RPE, SpO2, SBP, and DBP.

Discussion

- Please, turn topic 5 into 4.1 and rename it as Strengths and limitations;

- Please, provide a new topic 4.2 named Practical implications and future perspectives. In this topic, add the main (and also potential) implications of the present study for the readers, researchers, healthcare managers, and patients. Lastly, add possible next steps for this research field.

I also recommend checking the punctuation and the English writing once again. 

I am looking forward to the revised version of this manuscript.

Author Response

First of all, I appreciate the changes throughout the manuscript. There was a significant improvement, as well as a more readable version of this study for different types of readers. For me, this is essential in order to ensure a potential translation of this knowledge into a real practice scenario. 

Despite several of my suggestions were not taken into account, the overall comments were expectedly attended. Therefore, I recommend the publication since it is suitable for its main purpose and the journal's scope. However, some minor revision is required as the following comments:

Author response: My co-authors and I would like to thank you for a review of our work. We really appreciate your comments and will address them immediately below.

Introduction

- Please, merge the second and third paragraphs into only one;

Author response: Done as requested

- Please, add information about the ceiling effect as well in the paragraph about the learning effect and made the necessary changes to ensure clarity;

Author response: Thank you for the above suggestion. An information about the ceiling effect has been added in the learning effect paragraph for better clarity

(…) „Adsett et al. have shown that when the 6MWT is repeated at least twice, the final distance covered during the testing tends to from 301 meters in the first test to 313 meters in the second test (p < 0.001) [18].“ (…)

- Please, merge the two last paragraphs into only one. If possible, leave the primary and secondary aims in a new paragraph.

Author response: Thank you for the above suggestion. It was done as requested

Methods

- Lines 115-117: Were these variables collected independently from the study or not? Who collected them and when? It is also important to describe how these variables were achieved. 

Author response: Thank you for the comment

This info has been addressed in the manuscript:

..‘from the 9th of October to the 10th of November 2021 involving patients with DM2. (Line 81-83)’

‘The assessor was a qualified physiotherapist in the use of 6MWT before the start of the study testing. The same assessor carried out all assessments. After each assessment, the data were stored and presented in a spreadsheet for data analysis. ‘

Results

- Please, double-check the previous comments.

Author response: Thank you for the above suggestion. We have double-checked previous comments and provided the requested correction. The mean age was checked and revised to match the text and table. Flowchart (Figure 1) was mentioned in the text. Reporting results CB before of TL was checked and adjusted accordingly. Topics in the result section were deleted based on your suggestion.

Table 2

- Similar to HR, provide the necessary changes in table 2 for RPE, SpO2, SBP, and DBP.

Author response: Thank you for the above comment on the necessary changes in Table 2. We have corrected lines according to HR to get the similarity

Table 2. Evaluation of the differences of CB and TL 6MWT assessment.

Variable

CB 6MWT

TL 6MWT

Difference

T-test value

6MWT (Best Score) (m)

520.5 ± 80.3

544.8 ± 93.1

-24.3 ± 60.6

-1.8

HR(bpm)

Before

81.2 ± 14.5

77.9  ± 12.6

3.2 ± 11.7

1.2

After

100.8 ± 20.7

86.8 ± 16.1

14.0 ± 20.0

2.9a

Recovery

86.5 ± 15.8

83.3 ± 12.3

3.2 ± 12.0

1.2

RPE (After)

7.6 ± 2.8

7.3 ± 2.5

0.2 ± 1.1

1.0

SpO2 (%)

Before

97.4 ± 1.9

97.7 ± 1.3

-0.3 ± 1.7

-0.9

After

97.2 ± 1.9

97.8 ± 1.9

-0.6 ± 1.5

-1.6

SBP (mmHg)

Before

138.7 ± 15.4

131.0 ± 15.6

7.7 ± 14.4

2.3a

After

151.3 ± 23.9

144.8 ± 24.2

6.4 ± 15.7

1.7

DBP (mmHg)

Before

86.5 ± 10.9

80.6 ± 14.0

5.8 ± 9.7

2.6a

After

92.1 ± 10.5

86.2 ± 17.0

5.8 ± 15.2

1.6

a indicate significance (p < 0.05); Values are expressed as mean and standard deviation;
RPE, Rate of Perceived Exertion Scale (1-10)

Discussion

- Please, turn topic 5 into 4.1 and rename it as Strengths and limitations;

Author response: Done as requested

- Please, provide a new topic 4.2 named Practical implications and future perspectives. In this topic, add the main (and also potential) implications of the present study for the readers, researchers, healthcare managers, and patients. Lastly, add possible next steps for this research field.

Author response: Thank you for the above suggestions. We have included a new topic, “4.2. Practical implications and future perspectives,” with content on the implications of the present study for the readers, researchers, healthcare managers, and patients. The requested next steps for this research field were added at the end of the new topic.

(…) “ The study showed that remote TL 6MWT assessed outdoors in reasonable climate parameters can reflect CB 6MWT physical fitness indoors. The nature of the outdoor TL 6MWT is more reflective of real life compared to the conventional 6MWT, where the artifi-cial corridor environment is usually free of distractions. Hence an outdoor TL 6MWT could be an alternative measurement to conventional CB 6MWT in patients with DM2 for use in practice, especially when an indoor corridor is unavailable.

Furthermore, the results of this study will allow researchers and/or health care pro-fessionals to use conventional physical fitness CB 6MWT to predict outdoor physical fit-ness and conversely. In addition, research will enable healthcare professionals to use the 6MWT in rural patients or communities to measure physical fitness, prescribe individual-ized exercise activities, and monitor their effectiveness, which can optimize treatment out-comes and the utilization of preventive and rehabilitation interventions.

Future research should focus on studying TL in different outdoor conditions, such as different terrains, weather influences or summer and winter temperatures, wind or air pol-lution, and their effects on the physical performance of patients, for more generalizing the use of remote TL 6MWT.

Finally, using reliable and valid wearable sensors, such as sports watches, may con-tribute to a better and more objective assessment of the patient's hemodynamic character-istics during the TL 6MWT. The implementation of wearable-assisted TL in the future may play a vital role in the widespread provision of adequate medical care, even under unfa-vorable socioeconomic conditions, to a more significant part of the patient population. “ (…)

I also recommend checking the punctuation and the English writing once again. 

Author response: The manuscript was checked with a focus on punctuation and English writing.